# Inherited Hypertrabeculation? Genetic and Clinical Insights in Blood Relatives of Genetically Affected Left Ventricular Excessive Trabeculation Patients

**DOI:** 10.3390/life15020150

**Published:** 2025-01-22

**Authors:** Balázs Mester, Zoltán Lipták, Kristóf Attila Farkas-Sütő, Kinga Grebur, Flóra Klára Gyulánczi, Alexandra Fábián, Bálint András Fekete, Tamás Attila György, Csaba Bödör, Attila Kovács, Béla Merkely, Andrea Szűcs

**Affiliations:** 1Heart and Vascular Centre, Semmelweis University, 1085 Budapest, Hungary; mester.balazs@stud.semmelweis.hu (B.M.);; 2Department of Pathology and Experimental Cancer Research, Semmelweis University, 1085 Budapest, Hungary

**Keywords:** family health, cardiomyopathy screening, left ventricular excessive trabeculation, noncompaction, cardiology, echocardiography, genetics

## Abstract

Genetically determined left ventricular excessive trabeculation (LVET) has a wide clinical spectrum ranging from asymptomatic subjects to severe heart failure with arrhythmias and thromboembolic events. Unlike other cardiomyopathies, the relatives of LVET patients never reach the spotlight of guidelines and clinical practice, although these family members can be often affected by these conditions. Thus, we aimed to investigate the relatives of LVET by multidimensional analysis, such as genetic testing, ECG and cardiac ultrasound (ECHO). We included 55 blood relatives from the family of 18 LVET patients (male = 27, age = 44 ± 20.8y), who underwent anamnesis registration. With Sanger sequencing, the relatives were classified into genetically positive (GEN-pos) and unaffected (GEN-neg) subgroups. In addition to regular ECG parameters, Sokolow-Lyon Index (SLI) values were calculated. 2D ECHO images were analysed with TomTec Arena, evaluating LV volumetric, functional (EF) and strain parameters. Individuals were categorized into JENNI-pos and JENNI-neg morphological subgroups according to the Jenni LVET ECHO criteria. Family history showed frequent involvement (arrhythmia 61%, stroke 56%, syncope 39%, sudden cardiac death 28%, implanted device 28%), as well as personal anamnesis (subjective symptoms 75%, arrhythmias 44%). ECG and ECHO parameters were within the normal range. In terms of genetics, 78% of families and 38% of relatives carried the index mutation. LV_SLI and QT duration were lower in the GEN-pos group; ECHO parameters were comparable in the subgroups. Morphologically, 33% of the relatives met Jenni-LVET criteria were genetically affected and showed lower LV_EF values. The frequently found genetic, morphological and clinical involvement may indicate the importance of screening and, if necessary, regular follow-up of relatives in the genetically affected LVET population.

## 1. Introduction

Left ventricular hypertrabeculation is a rare morphological malformation, but increasingly diagnosed with the advent of imaging studies, affects primarily the apical region of the heart. It is caused by the varying growth rate of the compacted and the trabeculated layers, and may also be presented in the normal population [1]. However, if it fulfills the cardiac ultrasound (ECHO) [2] or the gold standard cardiac magnetic resonance (CMR) diagnostic criteria [3,4,5,6], it can be termed as left ventricular excessive trabeculation (LVET) [1,2,7].

In some circumstances, the asymptomatic primary LVET with preserved left ventricular ejection fraction (EF) progresses to noncompact cardiomyopathy due to the impairment of ventricular function, with heart failure symptoms, arrhythmias and stroke [8]. Thus, certain aspects of the relationship between morphology and clinical presentation still need to be clarified, in which risk stratification algorithms help to assess the involvement of individuals [9,10,11,12].

During this process, it is necessary to identify red flags [9,10,13,14]. These can be volumetric, functional and strain parameters measured by imaging modalities, depolarization and repolarization abnormalities on electrocardiogram (ECG) and the subject’s personal and family history with special attention to arrhythmias, hereditary cardiomyopathies and the presence of sudden cardiac death. Late gadolinium enhancement, right ventricular involvement and rotational abnormalities may have prognostic value as well [15,16,17].

Genetic testing is also included in the risk stratification, as several cases of LVET are genetically determined; moreover, individuals with pathogenic genetics and preserved left ventricular function may also present red flags [14]. According to the literature, these genes primarily affect sarcomere and sarcolemma structures such as Titin or Myosin resulting in morphological modification [18,19,20], but they are also associated with ion channel (RYR2) and signal transduction genes (NOTCH) that may determine arrhythmias [21,22,23]; furthermore, they may overlap with other cardiomyopathies [24]. 

Screening the severity and progression of subjects in our LVET registry frequently revealed that the population’s family had a distinct cumulative cardiological history, and the relatives often have similar symptoms. Since cardiological screening of relatives is not mentioned either in the guidelines or the related literature, or in everyday clinical practice, these potentially affected individuals remain undetected. However, affected families often ask for a cardiological screening.

Thus, we aimed to study the clinical characteristics and genetic background of families related to LVET subjects with pathogenic mutations (index subject) and to compare the families’ ECHO and ECG parameters within the genetic and morphology-determined subgroups to explore the effect of pathogenic genetics on the relatives.

## 2. Materials and Methods

### 2.1. The Study Populations

From our registry, we chose 18 genetically affected *index subjects* diagnosed with LVET, whose blood relatives underwent cardiological screening at our institution. The diagnosis was evaluated by CMR using the Petersen (noncompacted/compacted layer of the myocardium is ≥2.3) and Jacquier (noncompacted myocardial mass is more than 20% of the compacted) definitions [3,4,25,26]. Furthermore, these primary subjects had at least one pathogenic or likely pathogenic LVET-associated mutation according to the American College of Medical Genetics (ACMG) [27] databases (Frankin [28], Varsome [29], ClinVar [30]), which are listed in Table 1. The LVET association of mutations was determined using the OMIM [31,32] and ClinGen [33,34] databases. The baseline CMR and detailed genetic characteristics of the *index subjects* can be found in Appendix A.

For the study population, we enrolled the blood relatives of the index subjects (n = 55, male = 27) in the study, namely child, sibling, parent and grandparent. Exclusion criteria were if the subject did not agree with this investigation, or we were not able to examine the subject because of external factors (e.g., too young age). The average age was 44 ± 20.8 years (adult n = 49, child n = 6); the youngest was 7 and the oldest was 79 years.

During the research, data were treated anonymously, and only the examination leader knew the subject coding numbers. All studies were in accordance with the 1964 Helsinki Declaration and its amendments, as well as the applicable ethical standards. The ethical approval was validated by the Central Ethics Committee of Hungary; all participants received full information, which was confirmed by their signature.

### 2.2. Methods

All the 55 blood relatives underwent a questionnaire containing cardiovascular-focused personal and family history, an ECG scan, a cardiac ultrasound and genetic testing (Figure 1).

#### 2.2.1. Anamnestic Questionnaire

Family and personal history of the subjects was recorded using an anamnestic questionnaire containing both open-ended and closed questions. For the personal anamnesis, we gathered information about subjective symptoms like syncope, chest pain, dyspnoea and palpitations; previous cardiac history, e.g., documented arrhythmias, implanted devices such as pacemakers or implantable cardioverter defibrillators (ICDs), ischemic heart disease, cardiomyopathy; other comorbidities like hypertension, diabetes or stroke; medications and regular sport activities.

In the family history, we collected information about the presence of hereditary heart diseases, sudden cardiac death, syncope, arrhythmias, implanted devices, heart attack or stroke in the family of the study subjects. During the analysis, all the family members’ responses of one *index subject* were handled as one unit.

#### 2.2.2. ECG Scan

Subsequently, in the supine position, 12-lead ECG (HeartScreen 112 C-1, Innomed Medical, Budapest, Hungary) was performed at a paper speed of 25 mm/second; for the evaluation, the average of five beats was calculated. The frequency, duration and amplitude of the P and T waves and the duration of the PQ, QRS and QT periods were determined [35]. QTc was determined according to Bazett’s formula. From the QRS amplitude, left and right ventricular Sokolow–Lyon Index values (SLI_LV and SLI_RV) and the Cornell Voltage Criteria (CVC) were calculated [36,37]. The formulas and the normal values were determined in accordance with the American Heart Association [38].

The ECG scans were analysed by BaM and ZL.

#### 2.2.3. Cardiac Ultrasound

All subjects underwent transthoracic 2D ECHO, where ECG-gated images were recorded using GE Vivid E95 ultrasound machines with 4Vc transducers. During the cardiac ultrasound, parasternal short- and long-axis, and apical two-, three-, four-, and five-chamber images were recorded with left ventricular focus. Four subjects were excluded from the echocardiography analysis due to a non-optimal echo window or to other comorbidities that can affect cardiac function. The recordings were analysed using the Philips Ultrasound Workspace (version 6.0.0.0) software, where the volumetric and functional parameters were evaluated using the LV Autostrain module (Philips Ultrasound Workspace, TOMTEC Imaging Systems GmbH, Unterschleissheim, Germany). The module was applied to determine left ventricular end-diastolic volumes (LVEDVs), left ventricular end-systolic volumes (LVESVs), left ventricular stroke volumes (LVSVs) and LVEF parameters using end-systolic and end-diastolic endocardial contours of the left ventricle. The parameters were indexed to the body surface area (*i*), and normal values were determined with the European Association of Cardiovascular Imaging (EACVI) guidelines. The module is also able to perform speckle-tracking echocardiography (STE) so that the LV global longitudinal strain (GLS) was also determined from these contours.

To determine the amount of left ventricular trabecules, we used the Jenni LVET criteria, whereby the ratio of left ventricular trabeculated to compact muscle layer thickness was determined on SA end-systolic image (Jenni) [2]. When the noncompact ratio exceeded twice the thickness of the compacted muscle, the subject was considered Jenni-positive (JENNI-pos); any other cases were considered Jenni-negative (JENNI-neg) (Figure 2).

The ultrasound images were independently analysed by AS (7-year experience) and BaM (3-year experience).

#### 2.2.4. Genetic Testing

Automated extraction of genomic DNA from peripheral blood samples was performed using MagCore^®^ Plus II robotic bench-top workstation with MagCore^®^ Genomic DNA Whole Blood Kit 101 (RBC Bioscience Corp. New Taipei City, Taiwan). After DNA extraction, a focused genetic analysis was performed to indicate the primer mutation of index subjects affected with LVET.

Primer3Plus software (version 2.6.1) was used for designing the primers with a product size of 300–400 base pairs (primer length was 19–24 base pairs, Tm: 60 °C), and the position of the mutation was determined in the Ensembl database (human genome GRCh37). The sequence of the oligonucleotides used for polymerase chain reaction (PCR) amplification and direct Sanger sequencing is detailed in Appendix A. PCR amplification was performed using the AmpliTaq Gold™ 360 Master Mix (Thermo Fisher Scientific, Waltham, MA, USA), followed by digestion with ExoSAP-IT™ PCR Product Cleanup Reagent (Thermo Fisher Scientific, Waltham, MA, USA). For bidirectional Sanger sequencing, we used the BigDye^®^ Terminator v3.1 Cycle Sequencing chemistry (Thermo Fisher Scientific, Waltham, MA, USA) followed by the analysis of the terminated fragments on a 3500 Genetic Analyser (Thermo Fisher Scientific, Waltham, MA, USA). Sequences of AB1 files were analysed using the BioEdit (7.7) sequence alignment editor software. The above-mentioned procedures are in line with the MIQE guidelines.

Based on the results, family members were classified into a genetically positive (GEN-pos) and a genetically not-affected (GEN-neg) group. The genetical subgroups have been compared with an age- and sex-matched control group in Appendix A.

#### 2.2.5. Statistical Analysis

Continuous variables were described by means and standard deviations (SDs), and discrete values were defined as numbers of units and percentages. To explore the normal distribution, the Shapiro–Wilk test was used. To determine the differences in continuous ECHO and ECG variables between the genetical (GEN-pos and GEN-neg) and the morphological (JENNI-pos and JENNI-neg) subgroups, an independent *t*-test was used in normal distribution, and a Mann–Whitney U test was used in non-normal distribution. Comparisons among genetic subgroups and an age- and sex-matched control group in Appendix A were conducted using one-way analysis of variance (ANOVA), after which post hoc tests were applied between subgroups (Tukey’s post hoc test for normally distributed variables with equal variances, the Welch test with Games-Howell post hoc test for variables with unequal variances, and Kruskal–Wallis test with Bonferroni correction for non-normally distributed data). In anamnestic data, ECG characteristic Chi-square test and F-test were performed to compare the discrete variables across groups. Statistical analyses were performed using IBM SPSS Statistics (v28.0)

## 3. Results

The interobserver variabilities between the examiners in the ECG and ECHO analysis can be found in Appendix A.

### 3.1. Family and Personal Analysis

First, we analysed the cumulative history of the 18 families, and we frequently found positive cardiological involvement among them such as arrhythmia (61.1%), stroke (55.6%), syncope (38.9%), sudden cardiac death (27.8%) and pacemaker or ICD implantation (27.8%). Almost half of the families (eight families, 44.4%) had multiple positive findings in their family history, with three or four red flags of the previously listed conditions Table 2A.

Among the personal anamnesis of the 55 individuals, we highlighted the high prevalence of subjective symptoms (74.6%), the incidence of arrhythmias (43.6%) and hospitalization for cardiological reasons (18.2%). The cumulative anamnestic data can be found in Table 3, which is more detailed by person in Appendix A.

Regarding the total population, ECG and ECHO parameters were within the normal range. Notably, the ECG characteristics underscored a pronounced incidence of abnormalities (27.3%), of which interventricular conduction abnormalities were the most frequent (Appendix A).

### 3.2. Genetic Based Analysis

Based on the genetic samples 77,8% of the families had the index subject’s mutation among their relatives, and of 55 family members, 21 individuals (38.2%) carried the genetic mutation reported in the index subject (GEN-pos) and the remaining 34 subjects were assigned to group GEN-neg. The inherited variants and the number of mutation carriers in the families are shown in Table 2B. The genetic pedigrees of each affected family can be found in Appendix A. The genetic distribution by age and sex did not differ from the total population; namely, the genetic involvement was 33.3% of males, 42.9% of females, 33.3% of children, and 38.8% of adults. Regarding the subject history, GEN-pos and GEN-neg subgroups are compared in Table 4A.

During the analysis of the ECG recordings, the parameters were in the normal range and the SLI_LV and QT duration were significantly lower in the GEN-pos compared to the GEN-neg subgroup. The listed ECG abnormalities showed no correlation with the genetic involvement. Detailed ECG results can be found in Table 4B.

Regarding the ECHO evaluation, out of the GEN-pos subjects, 81.0% fulfilled the JENNI-pos criterion, and out of the GEN-neg subjects, only one was JENNI-pos. Upon comparing the genetic subgroups, the volumetric and functional parameters were within the normal range, exhibiting no significant differences as detailed in Table 4C.

The ECG and ECHO parameters were also compared with an age- and sex-matched control group; however, no significant differences were found. The comparison is presented in Appendix A.

### 3.3. Morphology Based Analysis

Cardiac morphology was determined using ultrasound scans, revealing that 32.73% of the total population met the JENNI-pos criterion, of whom all but one also carried the GEN-pos mutation. Comparing the anamnestic data between the JENNI-pos and the JENNI-neg subgroups yielded no differences (Table 5A).

The analysis of the ECG recordings showed comparable values, and the ECG characteristics did not differ between the two groups Table 5B.

Comparing the ECHO parameters between the subgroups, the measurements were found to be within the normal range; however, the EF was significantly lower in the JENNI-pos subgroup compared to JENNI-neg, as explained in Table 5C.

## 4. Discussion

In our study, we investigated the clinical, genetic and imaging involvements of blood relatives related to genetically affected LVET index subjects.

The variable morphological and clinical appearance of primary LVET, which does not represent a disease in itself, but can turn into noncompaction cardiomyopathy, is determined by a wide range of genetic mutations. Although some genes could be specific to LVET (e.g., NOTCH), the majority of variants involve sarcomere proteins such as Titin or Myosin, variants of which also occurred at high rates in our subject. However, these genes may overlap with other cardiomyopathies, such as DCM, HCM or even ACM. In addition to this genetic background, morphological signs may also overlap, hypertrabeculation appears in different cardiomyopathy, or a volumetric, dimensional or functional change occurs, which is often encountered in everyday clinical practice, and therefore diagnosis requires particular care [19,20,39,40,41,42,43].

This genetic background and heritability may help to explain why LVET is common in the families [43]. In our study, we found a carrier mutation in almost 80% of these families; moreover, half of them had multiple red flags. Meshkov NA et al., in a genetic landscape study of a Caucasian population with both good and reduced left ventricular function, found similarly high rates of familial mutation carrying. They also described that with a more frequent family history, a higher rate of pathogenic mutations and a more severe form of LVET occurred [44]. Despite the known familial accumulation, there is still limited literature on the clinical assessment regarding the families of subjects with LVET [1].

Although cardiomyopathies are known to be inherited in autosomal, X-linked or even mitochondrial patterns, less information is available on the inheritance of LVET: most cases refer to an autosomal dominant or X-linked pattern, and the mitochondrial inheritance is controversial [39,40,43,45,46,47]. Similarly, in our study, approximately half of the family members carried the mutation, as all index subjects were detected with the variant in heterozygous form, so this incidence suggests autosomal dominant inheritance also in our study population. The finding that less than half of the population carried the variant, of which not all were hypertrabeculated, is probably related to gene penetrance. Although there is limited information in the literature regarding this aspect of LVET, a Polish study reported that the disease was most often inherited in an autosomal dominant pattern, but the gene penetration was variable [48]. However, it is now well established in the similar genetic-based HCM that the penetrance, as a specific form of genotype-phenotype interaction, depends on the mutated gene, as Myosin light chain (MYL3) mutations had lower Troponin (TNNT2 and TNNT3), and Myosin heavy chain (MYH7) mutations had much higher penetrance [49].

In our clinical analysis, we found no difference in genetic involvement between child and adult populations, as well as males and females. Although LVET is often associated with males, besides our study, Meshkov NA et al. also described similar rates regarding age and sex. On the contrary, in other cardiomyopathies age- and sex-specific penetrance was described, although it may depend on the screening thoroughness and the age of initial presentation to the healthcare facilities [50,51,52,53].

### 4.1. Baseline Characteristics of the Total Population

Although the baseline ECG and ECHO characteristics of LVET subject families are currently not well explored in the literature, it is remarkable that all the ECHO and ECG parameters of the family members were within the normal range.

In addition, we found ECG abnormalities in several family members. Sanna DG. et al. stated that in adults and children with LVET, ECG abnormalities are quite common, so our finding is not unexpected. Furthermore, they claimed that these abnormalities are not specific enough to make a diagnosis on their own, but rather assist in decision making, which supports the fact that we did not see any group-specific ECG abnormalities in our genetic and morphological analyses as seen below [54].

### 4.2. Comparing GEN-Pos and GEN-Neg Subgroups

The genetic subgroups showed significant differences in ECG parameters between each other in the SLI_LV and QT duration. The elevated SLI, as a well-known marker of ventricular hypertrophy, is highly measurable and frequently used in other cardiomyopathies, mostly in HCM [48,55]. On the contrary, we found decreased SLI values in the GEN-pos subgroup; however, the role of the SLI in the case of LVET is still unexplored. Nevertheless, it is recognized that an apical-basal compact layer thickness gradient is present in LVET, and this myocardial gradient may result in reduced SLI [56]. Furthermore, it is also less known how the conducting Purkinje fibers act in the trabeculae questioning their electrical activity. Thus, in apical hypertrabeculation, the reduction in the compact muscle mass and the possible lower electrical activity together may result in decreased SLI_LV [1,48,55,57,58]. Moreover, the female population often has lower muscle mass values, since a higher proportion of women were in the GEN-pos group, which can be also the background of this difference [59,60]. Furthermore, the specificity and sensitivity of this parameter are controversial [61].

For the repolarization differences, namely the shorter QT duration seen on ECG, ion channel mutations may be responsible. The rapidly growing literature in LVET suggests that ion channel genes, notably the RYR2 or SCN5A, may develop an arrhythmic appearance in these subjects [22,62,63]. Besides the morphology-related sarcomere mutations, these ion channel mutations were also highly present in our population, but since there is less evidence about their pathogenicity yet, they are often categorized into a variant of uncertain significance (VUS). As we did not test the VUS mutations among the relatives, it is possible that these ion channel mutations were more likely presented in our GEN-neg subgroup accessing longer QT duration, while the GEN-pos subgroup inherited more morphology modifying mutations, generating a lower SLI value. 

Interestingly, in the ECHO comparison of the subgroups, we found no difference in volumetric, functional and strain parameters between genetically affected and unaffected subjects, which is in line with Grebur K et al.’s investigation using CMR [14]. This result may also be due to the penetrance, as the presence of a genetic mutation affecting morphology does not necessarily produce morphological changes in a certain patient or at the time of the sampling [49].

### 4.3. Comparing JENNI-Pos and JENNI-Neg Subgroups

When comparing according to the morphology, approximately one-third of the subjects fulfilled Jenni’s LVET diagnostic echocardiographic criteria, of whom almost all were genetically affected. They had significantly lower LV_EF values in ECHO compared to those without the morphology.

The subclinical LV_EF reduction seen on ECHO is consistent with CMR studies that investigated LVET subjects with preserved left ventricular EF. Regarding this subclinical change, Kiss A et al. stated that subjects with LVET morphology may have lower LV_EF compared to the normal population [64], similar to our ECHO study. Meanwhile, in a different CMR study, Zemrak F et al. found that during a 10-year follow-up, these subclinical EF changes did not worsen [65]. The changes in GLS values in LVET subjects are controversial: there are reports of a possible GLS decrease in LVET subjects, while other studies show no difference in GLS values [64,66]. Murphy R T et al. claimed that among relatives of LVET subjects, characteristics of DCM occurred more frequently than LVET features; however, our investigation did not support this finding [67].

### 4.4. Clinical Symptoms

The family members had cumulative clinical symptoms, with many cases of subjective symptoms, regardless of genetic background and morphology, which is in accordance with the findings of Grebur K et al.; interestingly, complaints also appeared in younger family members [14]. Regarding the family history of the relatives, many cases of arrhythmia and stroke are mentionable, as they are major risk factors in LVET [1,9].

In summary, the frequently appearing mutated variants in the families and the potentially related anamnestic, ECG and ECHO findings are remarkable and suggest that the follow-up of relatives of LVET subjects might also be of high importance. Further testing may help in risk stratification of LVET relatives, allowing us to detect and treat them in time if necessary and to prevent the development of noncompaction cardiomyopathy originating from the same genetic background.

## 5. Conclusions

In our study, we examined blood relatives of genetically affected LVET subjects. Approximately 80% of the families and 40% of the individuals carried the genetic mutations. The genetically affected and not affected subgroups showed no differences regarding the ECHO comparison and differed only in the ECG parameters. However, in the morphology, one-third of the subjects fulfilled Jenni’s LVET diagnostic echocardiographic criteria, of whom almost all were genetically affected and showed lower EF values. Regarding the clinical anamnesis, there were no differences between either subgroup.

While the current guidelines do not address clinical screening of LVET relatives, the genetic and morphological involvement in our results raises attention to the LVET family members in clinical practice. These findings highlight that risk stratification of the family members of LVET patients is recommendable, which should include cardiac ultrasound screening, complemented by additional examinations like genetic testing, CMR or long-term follow-up if necessary [1,52,53,68]. For more detailed information, further investigations with various conditions are recommended.

### Limitations

During the genetic testing of the relatives, we only tested the pathogenic mutations that the family members carried; however, many index subjects had other VUS mutations. The effect of these mutations in modifying the clinical presentation of each group cannot be excluded, but there is not enough available information to determine this outcome. Although the screening of these mutations among the relatives is not generally recommended during clinical practice [1], further genetic analysis including VUS genes could extend the results of our study.

Another limitation of genetic studies is the potential lack of sensitivity of the genetic tests used, which could have masked subtle differences between groups. This may improve in future research as wider genetic information is available.

Further and more accurate quantification of trabeculation would require a CMR scan, especially in the JENNI-pos group, which was not performed during the research.

In order to provide more accurate results from the subgroup analyses, a further increase in the number of family members may be required.

All family members who participated in the family screening were included in our study; therefore, the sex ratio in the subgroups was not necessarily identical. Investigating a larger population would further reduce the possible effect of different numbers of cases of each sex.

All subjects in the study population were selected from the Hungarian population and the studies were performed in the same clinic. A multi-ethnic and multicentric extension of the study may be necessary to achieve even more accurate results.

Our results mentioned above show this topic is worth further exploring, with advanced techniques like CMR, and we would like to address it in the future with multicentric examinations.

## Figures and Tables

**Figure 1 life-15-00150-f001:**
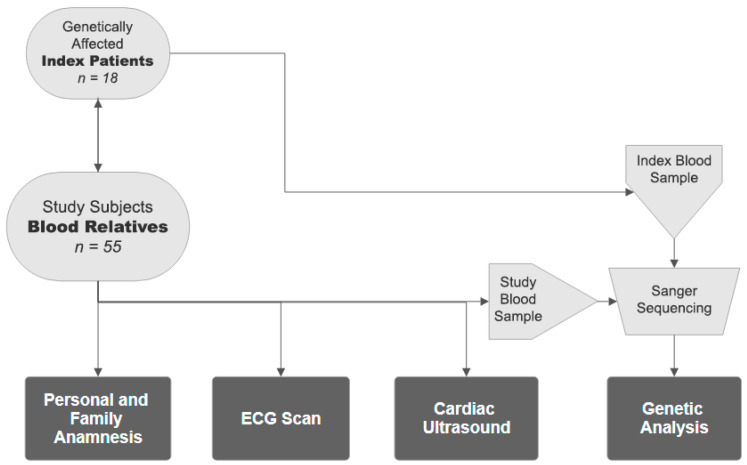
Screening model of study subjects. ECG: electrocardiogram, n: number of subjects.

**Figure 2 life-15-00150-f002:**
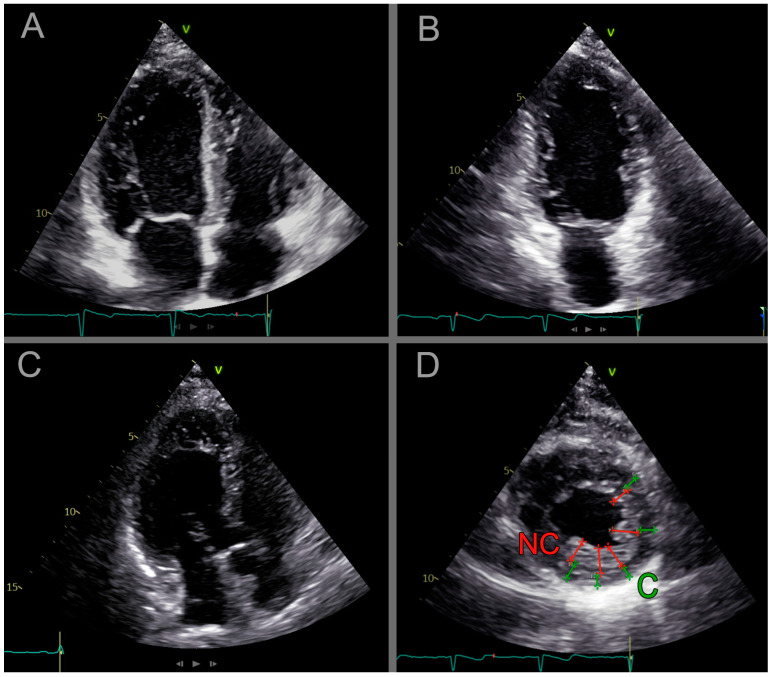
Cardiac ultrasound images of the blood relatives with hypertrabeculation. (**A**): Four-chamber view; (**B**): Two-chamber view; (**C**): Three-chamber view; (**D**): End-systolic short-axis view of the left ventricle; apical third. The arrows represent the Jenni criterion, where the red line shows the noncompacted (NC) and the green line shows the compacted (**C**) layer diameter and the ratio of NC/C > 2.

**Table 1 life-15-00150-t001:** The identified pathogenic or likely pathogenic gene mutations in the families based on the mutations of the *index patients* and their relation to cardiomyopathies according to the ClinGen [33,34] and OMIM [31,32] databases. More detailed information about mutations related to each index patient can be found in Appendix A.

Gene Symbol	Affected Protein	Relation to Cardiomyopathies and Arrhythmogenic Diseases						Number of Families
		LVET	HCM	DCM	ACM	Oth. Ar	Cong. CM	
TTN	Titin	+	+	+			+	9
MYH7	Myosin, Heavy Chain 7	+	+	+	+		+	2
MYBPC3	Myosin-Binding Protein C3	+	+	+	+			1
LMNA	Lamin A	*		+	+			1
DES	Desmin	*		+	+			1
RYR2	Ryanodine receptor 2	*	+		+	+		1
MYPN	Myopalladin		+	+			+	1
SCN5A	Sodium Voltage-Gated Channel Alpha Subunit 5	*		+	+	+		1
MIB1	MIB E3 Ubiquitin Protein Ligase 1	+		*				1
KCNQ1	Potassium Voltage-Gated Channel Subfamily Q Member 1	*	*			+		1

LVET: Left ventricular excessive trabeculation, HCM: Hypertrophic cardiomyopathy, DCM: dilatative cardiomyopathy, ACM: Arrhythmogenic cardiomyopathy, Oth. Ar: Other arrhythmogenic cardiac diseases, Cong. CM: Frequent mutation in congenital cardiomyopathies, +: Gene mutation is associated with the disease, *: not associated but occurring in the literature.

**Table 2 life-15-00150-t002:** The anamnestic (**A**) and genetic (**B**) details of the families. In section B, the genetic mutations of index patients, the presence of the mutation in the family and the number of affected family members in each family are presented.

Family Code Number	(A) Cumulative Family Anamnesis						(B) Family Genetics		
	Sudden Cardiac Death	Syncope	Arrhythmia	Pace-Maker or ICD	Stroke	Red Flags (n)	Affected Gene of the *Index Patient*	Presence of the Mutation in the Family	Number of Affected Family Members/Studied Family Members
LVET014						0	MYH7	no	0/2
LVET016	x	x	x		x	4	KCNQ1	yes	2/5
LVET031	x	x	x		x	4	MIB1	yes	1/5
LVET002		x	x	x	x	4	TTN	yes	1/3
LVET041						0	SCN5A	yes	1/1
LVET045			x	x		2	TTN	yes	1/1
LVET020						0	TTN	no	0/2
LVET006	x		x		x	3	TTN	yes	3/4
LVET023		x			x	2	MYPN	yes	2/2
LVET029	x	x	x			3	TTN	no	0/2
LVET004			x	x		2	RYR2, TTN	yes	2/3
LVET049		x	x		x	3	TTN	yes	1/4
LVET013		x	x		x	3	DES	no	0/7
LVET030				x	x	2	LMNA	yes	1/1
LVET040			x			1	TTN	yes	1/3
LVET025			x		x	2	MYBPC3	yes	2/5
LVET028	x			x	x	3	MYH7	yes	2/3
LVET046						0	TTN	yes	1/2
	Total (n)					Average		Total (n)	
	5	7	11	5	10	2.79		14	21/55
	Percentage (%)					Deviation		Percentage (%)	
	27.78%	38.89%	61.11%	27.78%	55.56%	1.50		77.78%	38.18%

LVET number: individual families, ICD: implantable cardioverter-defibrillator, TTN: Titin, MYH7: Myosin Heavy Chain 7, MYBPC3: Myosi- Binding Protein C3, LMNA: Lamin A, DES: Desmin, RYR2: Ryanodin receptor 2, MYPN: Myopalladin, SCN5A: Sodium Voltage-Gated Channel Alpha Subunit 5, MIB1: MIB E3 Ubiquitin Protein Ligase 1, KCNQ1: Potassium Voltage-Gated Channel Subfamily Q Member 1.

**Table 3 life-15-00150-t003:** The descriptive (**A**) and Personal anamnestic details (**B**) of the total family members.

**A**	**Total Family Members n = 55**	Age (average years, deviation years)	43	±20.8
		Sex (male n, male %)	27	49.09%
**B**	**Subjective Symptoms (n, %)**	Syncope	8	14.55%
		Chest pain	18	32.73%
		Dyspnoea	16	29.09%
		Palpitation	18	32.73%
		Total Subjective Symptoms	41	74.54%
	**Anamnestic Information (n, %)**	Documented Arrhythmia	15	27.27%
		Non-documented Arrhythmia	9	16.36%
		Cardiac hospitalization	10	18.18%
		Stroke	0	0.00%
		Ischemic disease	1	1.82%

Total Subjective Symptoms: family members, who had at least one subjective cardiac symptom.

**Table 4 life-15-00150-t004:** (**A**–**C**): Anamnestic (**A**), ECG (**B**) and cardiac ultrasound (**C**) comparison of the **genetic subgroups**.

**A**	**n**	**Syncope**	**Chest Pain**	**Dyspnoe**	**Palpitation**	**Documented Arrhythmia**		**Non-Documented Arrhythmia**	**Cardiac Hospitalization**		**B**	**ECG Abnormality**			
**Anamnesis**	GEN-pos (21)	3	7	7	7	6		6	2		**ECG Characteristics**	7			
	GEN-neg (34)	5	11	9	11	3		10	8			8			
	*p*	*0.966*	*0.940*	*0.586*	*0.940*	*0.743*		*0.947*	*0.191*			*p*	*0.428*	*r*	*0.107*
**B**	**n**	**Frq. (b/m)**	**P dur.** **(ms)**	**P amp. (mV)**	**PQ (ms)**	**QRS (ms)**	**SLI_LV (mV)**	**SLI_RV (mV)**	**CVC (mV)**	**QT** **(ms)**	**QTc (Bazett) (ms)**	**T dur.** **(ms)**		**T amp. (mV)**	
**ECG**	GEN-pos (21)	77.2 ± 14.6	97.4 ± 15.9	0.11 ± 0.03	155.8 ± 18.3	102.1 ± 16.9	15.7 ± 5.4	4.6 ± 4.1	12.2 ± 7.1	368.4 ± 26.0	414.4 ± 32.7	170.3 ± 19.8		0.3 ± 0.2	
	GEN-neg (34)	71.7 ± 12.1	97.6 ± 13.5	0.12 ± 0.03	151.4 ± 24.8	99.8 ± 14.0	20.1 ± 7.1	4.0 ± 7.0	11.2 ± 5.1	382.3 ± 32.4	415.9 ± 44.1	172.6 ± 24.8		0.3 ± 0.2	
	*p*	*0.10*	*0.503*	*0.019 **	*0.506*	*0.295*	*<0.001 **	*0.445*	*0.687*	*0.199*	*0.131*	*0.890*		*0.488*	
**C**	**n**	**JENNI-poz** ** *n (%)* **		**LV_EDV*i*** **(mL/m^2^)**		**LV_ESV*i*** **(mL/m^2^)**		**LV_SV*i*** **(mL/m^2^)**		**LV_EF** **(%)**		**LV_GLS** **(%)**			
**ECHO**	GEN-pos (21)	17 (81.0%)		58.0 ± 18.5		25.2 ± 8.7		33.3 ± 9.6		57.2 ± 3.8		−19.9 ± 2.4			
	GEN-neg (30)	1 (3.3%)		59.6 ± 11.4		26.0 ± 6.7		35.6 ± 6.9		58.6 ± 4.1		−20.1 ± 2.5			
	*p*	*<0.001 **		*0.228*		*0.379*		*0.168*		*0.120*		*0.359*			

GEN-pos: family members who carry the index patient’s mutation, GEN-neg: family members who do not carry the index patient’s mutation, ECHO: cardiac ultrasound, n: total study population number, Frq (b/m).: frequency (beat per minute), P dur.: P duration, P amp.: P amplitude, SLI_LV: left ventricular Sokolow index, SLI_RV: right ventricular Sokolow index, CVC: Cornell Voltage Criteria index, QTc (Bazett): Bazett’s formula corrected QT time, T amp.: T amplitude, T dur.: T duration, T amp.: T amplitude, LV_EDV: left ventricular end-diastolic volume, LV_ESV: left ventricular end-systolic volume LV_SV: left ventricular stroke volume, LV_EF: left ventricular ejection fraction, LV_GLS: left ventricular global longitudinal strain, *i*: body surface area indexed parameter, *p*: value of significance, *: *p* < 0.05.

**Table 5 life-15-00150-t005:** (**A**–**C**): Anamnestic (**A**), ECG (**B**) and cardiac ultrasound (**C**) comparison of the **morphological subgroups** according to the Jenni criteria.

**A**	**n**	**Syncope**	**Chest pain**	**Dyspnoe**	**Palpitation**	**Documented Arrhythmia**		**Non-documented Arrhythmia**	**Cardiac Hospitalization**		**B**	**ECG Abnormality**			
**Anamnesis**	JENNI-pos (18)	3	6	5	6	6		4	3		**ECG Characteristics**	5			
	JENNI-neg (37)	5	12	11	12	3		12	7			10			
	*p*	*0.756*	*0.947*	*0.881*	*0.947*	*0.966*		*0.434*	*0.839*			*p*	*0.953*	*r*	*0.08*
**B**	**n**	**Frq. (b/m)**	**P dur.** **(ms)**	**P amp. (mV)**	**PQ (ms)**	**QRS (ms)**	**SLI_LV (mV)**	**SLI_RV (mV)**	**Cornell (mV)**	**QT** **(ms)**	**QTc (Bazett) (ms)**	**T dur.** **(ms)**		**T amp. (mV)**	
**ECG**	JENNI-pos (18)	76.8 ± 15.7	99.6 ± 15.3	0.11 ± 0.03	153.9 ± 19.5	101.9 ± 13.7	17.6 ± 5.1	4.4 ± 3.8	12.8 ± 6.4	368.0 ± 27.6	412.6 ± 34.6	169.3 ± 15.6		0.3 ± 0.2	
	JENNI-neg (37)	72.4 ± 12.3	95.9 ± 14.1	0.12 ± 0.02	154.1 ± 25.1	99.1 ± 16.1	18.5 ± 7.4	4.0 ± 2.7	10.3 ± 4.9	379.5 ± 29.1	413.8 ± 42.3	172.1 ± 15.6		0.3 ± 0.2	
	*p*	*0.174*	*0.197*	*0.213*	*0.49*	*0.315*	*0.399*	*0.812*	*0.056*	*0.067*	*0.914*	*0.309*		*0.953*	
**C**	**n**	**GEN-poz** ** *n (%)* **		**GEN-neg** ** *n (%)* **		**LV_EDV*i*** **(mL/m^2^)**		**LV_ESV*i*** **(mL/m^2^)**	**LV_SV*i*** **(mL/m^2^)**	**LV_EF** **(%)**		**LV_GLS** **(%)**			
**ECHO**	JENNI-pos (18)	17		1		60.9 ± 17.9		26.7 ± 8.5	34.2 ± 9.6	56.2 ± 7.7		−20.0 ± 2.5			
	JENNI-neg (33)	4		29		57.9 ± 12.5		25.1 ± 6.9	34.9 ± 7.3	59.0 ± 4.3		−20.0 ± 2.4			
	*p*	*<0.001 **		*<0.001 **		*0.844*		*0.568*	*0.379*	*0.010 **		*0.492*			

JENNI-pos: fulfilled the Jenni LVET diagnostic ultrasound criteria, JENNI-pos: do not met the Jenni LVET diagnostic ultrasound criteria, ECHO: cardiac ultrasound, GEN-pos: carries the index patient’s mutation, GEN-neg: mutation was not detected, n: total study population number, Frq. (b/m): frequency (beat per minute), P dur.: P duration, P amp.: P amplitude, SLI_LV: left ventricular Sokolow index, SLI_RV: right ventricular Sokolow index, CVC: Cornell Voltage Criteria index, QTc (Bazett): Bazett’s formula corrected QT time, T amp.: T amplitude, T dur.: T duration, T amp.: T amplitude, LV_EDV: left ventricular end-diastolic volume, LV_ESV: left ventricular end-systolic volume LV_SV: left ventricular stroke volume, LV_EF: left ventricular ejection fraction, LV_GLS: left ventricular global longitudinal strain, *i*: body surface area indexed parameter., senz.: sensitivity, spec.: specificity, *p*: value of significance, *: *p* < 0.05.

## Data Availability

The original data presented in this study are included in the article/Appendix A. Further inquiries can be request from the corresponding author due to ethical and privacy reasons.

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
