# Peer review of "Inherited Hypertrabeculation? Genetic and Clinical Insights in Blood Relatives of Genetically Affected Left Ventricular Excessive Trabeculation Patients"

_life, 2025, doi:10.3390/life15020150_

Round 1
Reviewer 1 Report
Comments and Suggestions for Authors
We thank the authors for their efforts to publish their experience with their large cohort of patients with LVH and their relatives.
I have 5 critics:
1 - Tables 4 and 5 are well done.
But while reading them, I keep asking myself a question without finding the answer in the text: why in the case (Gen neg/Jenny neg) of table 5 there are only 27 people reported and not 33?)
2 - The central illustration
I did not understand why you did not present your results in a classical contingency table with 2 variables GEN+/- and Morpho+/-, which is much more classical and understandable. But perhaps you have good reasons for doing so. In this case, please explain your reasons to me?
3 lines 206 - 208 you state "Regarding the total population, ECG and ECHO parameters were within the normal
range. Notably, the ECG characteristics underscored a pronounced incidence of intra-
ventricular conduction abnormalities (27.3%) Supplementary Material 2 Table 3." But in Table 3 you state that the total number of abnormal features on the ECG (percentage of family member's ECG) is 27.3%. Furthermore, for the echo data, the reported mean LV EF is 58.29 with a standard deviation of 4.22. This means that in the population of 55 relatives, some of them are in the range of mild abnormality (LV-EF 41 - 53%).
And you write it again, lines 297-98
4 - Discussion The overlap between currently 2 relatively distinct entities, LVET and LV non-compaction cardiomyopathy, is not brought into the discussion. Although your results greatly accentuate this overlap, the genes currently identified are the same. Please include this point in the discussion.
5 It is a pity that the population of gen+ or gen- relatives was not compared to a cohort of the general population matched by age and sex. By not doing this, you cannot tell if relatives have more clinical events than the general population.
It cannot be excluded that a lack of sensitivity of the genetic tests used could mask potential differences
Author Response
We would like to thank Reviewer 1 for the careful review and practical questions, which have greatly enhanced the clarity and rigor of our work. All insights are deeply appreciated. Please find below the replies for the recommendations.
- (Question 1)
Tables 4 and 5 are well done.
But while reading them, I keep asking myself a question without finding the answer in the text: why in the case (Gen neg/Jenny neg) of table 5 there are only 27 people reported and not 33?)
The Reviewer's first question concerned Table 5, in which he noted smaller number of cases in the (Jenny neg / Gen neg) group than it would be expected. This occurred because few subjects were excluded from the ECHO analysis due to technical reasons (e.g. poor echo window) as it was detailed in the Methods section, lines 146-148.
- (Question 2)
The central illustration
I did not understand why you did not present your results in a classical contingency table with 2 variables GEN+/- and Morpho+/-, which is much more classical and understandable. But perhaps you have good reasons for doing so. In this case, please explain your reasons to me?
Thanks to Reviewer for a very practical and thoughtful question. After studying the central illustrations of the journal, we found that they tended to be more of a graphical presentation with flowcharts for easier understanding. As we would like to meet the expectations of the journal, we have created the illustrations with a similar approach.
- (Question 3)
In lines 206 - 208 you state "Regarding the total population, ECG and ECHO parameters were within the normal range. Notably, the ECG characteristics underscored a pronounced incidence of intraventricular conduction abnormalities (27.3%) Supplementary Material 2 Table 3." But in Table 3 you state that the total number of abnormal features on the ECG (percentage of family member's ECG) is 27.3%. Furthermore, for the echo data, the reported mean LV EF is 58.29 with a standard deviation of 4.22. This means that in the population of 55 relatives, some of them are in the range of mild abnormality (LV-EF 41 - 53%).
And you write it again, lines 297-98
Thank Reviewer for the comment! In terms of ECG: the metric parameters (durations and amplitudes) of the ECG were found to be within normal range, however, the morphological appearance of the ECG may show abnormal features, such as intraventricular conduction abnormalities.
We would like to apologies for the misinterpretation of the Result’s text addressed to Supplementary Material 2 Table 3, namely the patients had not only conduction abnormalities but arrhythmias as well. Accordingly, we have modified the text of lines 230-233.
In addition, in the Anamnestic information section of Table 3 in the main document, the ECG abnormality row has been removed, since all anamnestic ECG abnormalities were included in the Documented or Non-documented arrhythmia rows.
As for the ECHO parameters: According to echo guidelines [1; 2], patients with EF 58.29 ± 4.22 % still falls within the normal range.
- (Question 4)
Discussion The overlap between currently 2 relatively distinct entities, LVET and LV non-compaction cardiomyopathy, is not brought into the discussion. Although your results greatly accentuate this overlap, the genes currently identified are the same. Please include this point in the discussion.
Thank the Reviewer for the foresighted and important comment on the appearance of LVET, which ranges from the asymptomatic person with good left ventricular function to the patient with cardiomyopathy and heart failure, a spectrum we have also emphasize in the Introduction with a similar concept.
The significance of our present study is precisely to screen for asymptomatic genetic carriers in time to focus on them and prevent them from developing heart failure.
Although the importance of this has been mentioned in the Discussion section, it has not been emphasised enough, so we have highlighted it as requested by the Reviewer. See lines 298-300.
For further clarification, the end of the discussion has also been modified, lines 400-405:
- (Question 5)
It is a pity that the population of gen+ or gen- relatives was not compared to a cohort of the general population matched by age and sex. By not doing this, you cannot tell if relatives have more clinical events than the general population.
It cannot be excluded that a lack of sensitivity of the genetic tests used could mask potential differences.
We thank the Reviewer for the useful insight on the comparison of genetic subgroups with a control group. Previously we performed a comparison of ECG and ECHO parameters with an age- and sex-matched control group, but found no significantly outstanding results, so for the sake of clarity we have not included it in the manuscript. However, for the sake of completeness, we have included these comparisons in Supplementary Material 2 Figure 2 and 3, furthermore we updated the Methods (lines 192-194 and 201-207) and Results (lines 254-256) sections.
However, we did not compare the subjects with the control group in terms of clinical symptoms, because the control group included subjects only without complaints and symptoms. Accordingly, we did not claim in the article that the subjects had more clinical events than the general population.
It is worth noting that, although no statistics were carried out, it is visible that family members have more complaints compared to controls with no complaints or symptoms at all.
Regarding the „lack of sensitivity of the genetic tests”, we have also addressed this point in the study's limitations section (lines 429-431)
We are deeply grateful for the Reviewer thorough revision and thoughtful comments, which have strengthened our manuscript. If any further questions or concerns remained, please do not hesitate to reach out to us.
[1] Lang, R. M., Badano, L. P., Mor-Avi, V., Afilalo, J., Armstrong, A., Ernande, L., Flachskampf, F. A., Foster, E., Goldstein, S. A., Kuznetsova, T., Lancellotti, P., Muraru, D., Picard, M. H., Rietzschel, E. R., Rudski, L., Spencer, K. T., Tsang, W., & Voigt, J. U. (2015). Recommendations for cardiac chamber quantification by echocardiography in adults: an update from the American Society of Echocardiography and the European Association of Cardiovascular Imaging. European heart journal. Cardiovascular Imaging, 16(3), 233–270. https://doi.org/10.1093/ehjci/jev014
[2] https://www.heart.org/en/health-topics/heart-failure/diagnosing-heart-failure/ejection-fraction-heart-failure-measurement
Reviewer 2 Report
Comments and Suggestions for Authors
The study includes multiple dimensions of analysis (e.g., genetic testing, ECG, echocardiography, and family history). The inclusion and exclusion criteria, tools, and statistical methods are clearly described. The study addresses a relatively underexplored area—Left Ventricular Excessive Trabeculation (LVET) in family members of affected individuals.
However, based on a review of the provided document, here are key points for critique and suggestions for improvement to enhance the study's clarity, rigor, and impact:
0. The abstract is informative but could better emphasize the study's clinical significance and potential contributions. Keywords could be more diverse to capture a broader audience interested in genetics, cardiology, and family health.
1. While the objectives are outlined, the research questions could be framed more precisely to highlight the novelty of the study. For example, focus explicitly on gaps addressed by this study compared to prior research.
2.Tables and figures, although comprehensive, are dense and may overwhelm the reader. Summarizing key findings graphically (e.g., infographics or simplified flowcharts) could enhance understanding. However, some descriptions in the results are repetitive.
3. Acknowledge the potential bias from excluding variants of uncertain significance (VUS) from the genetic analysis. Consider integrating more insights on genotype-phenotype correlations, especially how they compare to findings in other populations.
4.While the statistical methods are appropriate, some comparisons (e.g., GEN-pos vs. GEN-neg, JENNI-pos vs. JENNI-neg) could benefit from multivariate analysis to adjust for potential confounders like age, sex, and lifestyle factors.
5.The study cites existing research well, but it could provide more critical comparisons to demonstrate how findings align or diverge from prior studies.
The discussion would benefit from a more detailed exploration of how the findings could impact clinical screening guidelines or genetic counseling practices.
6.The conclusion mentions the need for family screening but could be strengthened with actionable recommendations for clinical practice or future research.
7.While limitations are discussed, the study could explore ways to address them in future research, such as including multi-ethnic cohorts or using advanced imaging techniques like CMR in all participants.
Author Response
We are very grateful for the Reviewer's work in reviewing and evaluating our manuscript, and further raising the quality of the material with valuable and useful insights. Please find our answers below.
- (Question 0)
The abstract is informative but could better emphasize the study's clinical significance and potential contributions. Keywords could be more diverse to capture a broader audience interested in genetics, cardiology, and family health.
The Reviewer had a great idea to emphasise the message in the abstract, so we highlighted the main messages in the opening and closing thoughts of the abstract as requested (lines 14-17 and 32-33).
We also modified the keywords to reach a wider audience (lines 35-36). We added “Cardiology”, modified “Family Screening” to “Family Health”, further included “Cardiomyopathy Screening”, and removed “Risk Stratification”.
- (Question 1)
While the objectives are outlined, the research questions could be framed more precisely to highlight the novelty of the study. For example, focus explicitly on gaps addressed by this study compared to prior research.
Thank the Reviewer for the valuable suggestion. We appreciate your emphasis on framing the research questions more precisely to highlight the study's novelty. We have revised this section to provide clearer articulation of the research gaps addressed, ensuring better alignment with the study’s objectives and enhancing the reader’s understanding of its contributions, in lines 74-79.
- (Question 2)
Tables and figures, although comprehensive, are dense and may overwhelm the reader. Summarizing key findings graphically (e.g., infographics or simplified flowcharts) could enhance understanding. However, some descriptions in the results are repetitive.
We appreciate the Reviewer’s insightful comment on the comprehensive presentation of our tables and figures. Our central illustration was designed to present the base of our results in a more visual form, aiming to provide a concise overview of the key findings. However, simplifying the tables would lead to a loss of valuable data or altering the original integrity, that we believe is crucial for readers seeking a comprehensive understanding. With that said, if it would be helpful, we could try to create an additional figure or infographic to be included in the supplementary materials.
Regarding the repetitiveness in the descriptions of the results, we acknowledge that some data from the tables are also detailed in the text of the results to ensure the critical information is emphasized for the reader. Nevertheless, we have reviewed the full Results section and found no accidental repetition in the text.
- (Question 3)
Acknowledge the potential bias from excluding variants of uncertain significance (VUS) from the genetic analysis. Consider integrating more insights on genotype-phenotype correlations, especially how they compare to findings in other populations.
Thank the Reviewer for the admission, we have also considered the potential bias introduced by excluding variants of uncertain significance (VUS) from the genetic analysis, but because current guidelines neither do not recommend including VUS in such analyses, we also did not include them in the examination. Additionally, the significant differences observed between groups when excluding VUS strengthen the robustness of our findings more than if VUS were retained. We have acknowledged this limitation and included it in Limitation section (lines 424-425).
Regarding genotype-phenotype correlations, we have expanded the Discussion section, reinforcing the idea at multiple points and providing a comprehensive review of the relevant literature to contextualize our findings (lines 298-300, 303-307 and 326).
- (Question 4)
While the statistical methods are appropriate, some comparisons (e.g., GEN-pos vs. GEN-neg, JENNI-pos vs. JENNI-neg) could benefit from multivariate analysis to adjust for potential confounders like age, sex, and lifestyle factors.
Thank the Reviewer for the insightful comment regarding the inclusion of sub-analysis to adjust for potential confounders such as age, sex, and lifestyle factors. We have considered this previously and addressed the genetic aspect detailed in lines 243-245.
During the evaluation we have performed age- and sex-adjusted statistical analyses, which we finally did not include in Results for some reasons: On one hand, the non-significant results did not enhance the interpretation of our data, moreover the more robust volume of information and tables would potentially reduce the transparency and clarity of the article. On the other hand, the division into such subgroups resulted in low sample sizes making statistical evaluation challenging.
To ensure completeness, we have included this data in a separate word document to the response letter for your review and we are happy to discuss further if needed. Please see the attachment.
- (Question 5)
The study cites existing research well, but it could provide more critical comparisons to demonstrate how findings align or diverge from prior studies.
The discussion would benefit from a more detailed exploration of how the findings could impact clinical screening guidelines or genetic counseling practices.
Thank the Reviewer for the thoughtful feedback. We have revised the discussion section and incorporated some relevant indicators (see in Discussion section) to include more critical comparisons with prior studies, highlighting how our findings align or diverge. Additionally, we expanded on the potential impact of our results on clinical practices, at the end of the Discussion section, lines 400-405.
- (Question 6)
The conclusion mentions the need for family screening but could be strengthened with actionable recommendations for clinical practice or future research.
We thank the Reviewer for the helpful comment, we have modified the conclusion to further highlight the clinical recommendation that follows from the research, as well as further investigational possibilities, detailed in lines 414-420.
- (Question 7)
While limitations are discussed, the study could explore ways to address them in future research, such as including multi-ethnic cohorts or using advanced imaging techniques like CMR in all participants.
We welcome the Reviewer's very helpful thoughts; we have also considered these in the Limitations section. However, to further strengthen this idea, we have added the following lines to the Limitations: lines 443-445.
Thank the Reviewer once again for the careful evaluation of our manuscript. The constructive feedback and suggestions have been invaluable in enhancing the clarity and quality of our study. If the Reviewer have any additional questions or thoughts, we are welcome to address them.

Reviewer 3 Report
Comments and Suggestions for Authors
The authors present an interesting study in which a distinct genetic population i.e. a family that has a history/high prevalence of left ventricular excessive trabeculation (LVET) is profiled in order to determine/identify risk factors related to the absence/presence of particular genes and how they correlate to clinical manifestations associated with such. Overall, the authors suggest based on their data that certain clinical manifestations agreed with the occurrence of certain genes and may act as representative ‘signs’ for cardiac problems, where as a lower association with other recognisable clinical signs are perhaps less reliable.
In reviewing the manuscript I made a couple of observations. The following should be considered by the authors when preparing a suitable revision.
1. Were the primers tested to ensure they complied with MIQE guidelines criteria?
2. The labelling/legend of the tables could be improved upon in terms of clarifying what certain labels/numbers refer to. For example, in Table 3 do the numbers with subjective symptoms refer to those in the group of 55, or other?
3. The formatting of Table 4 and 5 could be improved upon with the sizing of columns/rows making it difficult to read labels/headings.
4. In order for some p values to stand out it might be an idea to place a ‘*’ or use another means to make the values stand out in the tables etc. where data analyses are performed.
Author Response
We sincerely appreciate the Reviewer’s thorough evaluation of our manuscript and their valuable insights, which have significantly contributed to enhancing the quality of the material. Our responses to the questions are provided below.
1) Were the primers tested to ensure they complied with MIQE guidelines criteria?
Thanks for the Reviewer's question! The Genetic testing was performed by the Department of Pathology and Experimental Cancer Research, Semmelweis University. The institute is involved in genetic and molecular biology studies not only in research but also at clinical diagnostic level, which are thus performed systematically in compliance with the MIQE guidelines. DNAs were extracted from blood samples using the MagCore Plus II workstation, primers were designed using the Primer3Plus software. Mutation positions were determined using the Ensembl database using the Grc37 human reference genome. The use of these databases allowed the reproducibility of the data. These methods can be found in detail at the Method section – Genetic testing subsection, and in addition we included the fulfilment of MIQE guidelines in lines 189-190.
2) The labelling/legend of the tables could be improved upon in terms of clarifying what certain labels/numbers refer to. For example, in Table 3 do the numbers with subjective symptoms refer to those in the group of 55, or other?
Thank Reviewer for the comment regarding the clarity of table and figure legends. We have reviewed the titles and legends of all tables and figures and amended them as necessary to improve clarity. For example, in Table 3 we tought of the total population, that included the number of the blood relatives and added the „total” adjactive to the legend. Further corrections were also made in most of the tables and figures, and their legends.
3) The formatting of Table 4 and 5 could be improved upon with the sizing of columns/rows making it difficult to read labels/headings.
We appreciate the Reviewer’s feedback on the formatting of Tables 4 and 5. Originally, these results were designed as horizontal tables, but due to the journal’s formatting requirements, the editorial staff reformatted them into a vertical layout, which may have affected their dimensions and readability. In addition, we have rounded up to one decimal place to make the data more understandable. Thank Reviewer for bringing this to our attention; we will carefully address these formatting issues during the finalization process to enhance their readability.
4) In order for some p values to stand out it might be an idea to place a ‘*’ or use another means to make the values stand out in the tables etc. where data analyses are performed.
Thank Reviewer for the suggestion regarding the presentation of p-values. To ensure important values stand out, we have added (*) marks where necessary and updated the descriptions in the table legends to reflect this change.
We sincerely appreciate the Reviewer for the thoughtful review and questions, which has significantly contributed to improving the quality of our work. If the Reviewer have any further questions or suggestions, please feel free to share them with us.
Round 2
Reviewer 1 Report
Comments and Suggestions for Authors
Thank you for all the answers given
Author Response
We thank the Reviewer for the work and attention given to our research during the revision of the manuscript and the responses. The suggestions and questions have greatly improved the precision of the document. If any further questions arise in the future, we will be happy to address them.
Reviewer 2 Report
Comments and Suggestions for Authors
Accept in present form
Author Response
We thank the Reviewer for the careful review and evaluation of the manuscript and the replies, the questions and insights raised have greatly improved the quality of the document. If any questions arise in the future, we will be happy to answer them.